# Clinical Utility of SARS-CoV-2 Antibody Titer Multiplied by Binding Avidity of Receptor-Binding Domain (RBD) in Monitoring Protective Immunity and Clinical Severity

**DOI:** 10.3390/v15081662

**Published:** 2023-07-30

**Authors:** Etsuhisa Takahashi, Takako Sawabuchi, Tetsuya Homma, Yosuke Fukuda, Hironori Sagara, Takeshi Kinjo, Kaori Fujita, Shigeru Suga, Takashi Kimoto, Satoko Sakai, Keiko Kameda, Hiroshi Kido

**Affiliations:** 1Division of Enzyme Chemistry, Institute for Enzyme Research, Tokushima University, Tokushima 770-8503, Japan; etaka.tokushima.u@gmail.com (E.T.); sawabuchi.takako@tokushima-u.ac.jp (T.S.); kimoto@tokushima-u.ac.jp (T.K.); kameda.k@tokushima-u.ac.jp (K.K.); 2Division of Respiratory Medicine & Allergology, Showa University School of Medicine, Tokyo 142-8666, Japan; oldham726@yahoo.co.jp (T.H.); y.f.0423@med.showa-u.ac.jp (Y.F.); sagarah@med.showa-u.ac.jo (H.S.); 3First Department of Internal Medicine, Division of Infectious, Respiratory, and Digestive Medicine, University of the Ryukyu Graduate School of Medicine, Okinawa 903-0215, Japan; t_kinjo@med.u-ryukyu.ac.jp; 4Division of Pulmonary Medicine, National Hospital Organization Okinawa National Hospital, Okinawa 901-2214, Japan; fujita.kaori.ea@mail.hosp.go.jp; 5National Hospital Organization Mie National Hospital, Mie 514-0125, Japan; suga.shigeru.ke@mail.hosp.go.jp

**Keywords:** SARS-CoV-2, mRNA vaccine, protective immunity maturation, RBD binding avidity, viral neutralization, patient severity

## Abstract

Conventional serum antibody titer, which expresses antibody level, does not provide antigen binding avidity of the variable region of the antibody, which is essential for the defense response to infection. Here, we quantified anti-SARS-CoV-2 antibody binding avidity to the receptor-binding domain (RBD) by competitive binding-inhibition activity (IC50) between SARS-CoV-2 S1 antigen immobilized on the DCP microarray and various RBD doses added to serum and expressed as 1/IC50 nM. The binding avidity analyzed under equilibrium conditions of antigen–antibody binding reaction is different from the avidity index measured with the chaotropic agent, such as urea, under nonequilibrium and short-time conditions. Quantitative determination of the infection-protection potential of antibodies was assessed by ABAT (antigen binding avidity antibody titer), which was calculated by the quantity (level) × quality (binding avidity) of antibodies. The binding avidity correlated strongly (*r* = 0.811) with cell-based virus-neutralizing activity. Maturation of the protective antibody induced by repeated vaccinations or SARS-CoV-2 infection was classified into three categories of ABAT, such as an initial, low, and high ABAT. Antibody maturity correlated with the clinical severity of COVID-19. Once a mature high binding avidity was achieved, it was maintained for at least 6–8 months regardless of the subsequent change in the antibody levels.

## 1. Introduction

SARS-CoV-2 infection is triggered by the fusion of the viral membrane with the plasma membrane of the target cells, which is initiated by the binding of the receptor-binding domain (RBD) of the viral spike protein S1 to the angiotensin-converting enzyme 2 (ACE2) receptor on the cell membrane [1,2,3,4]. The virus-neutralizing activity of antibodies involved in protection against infection depends on the factors that can compete with RBD/ACE2 binding: one is the quantity of antiviral spike protein S1 antibodies and the other is the RBD binding-affinity/avidity strength of the antibodies [5,6].

Affinity indicates the binding interaction of a monoclonal antibody that binds to a single antigen epitope, and avidity indicates the sum of the binding affinities of each polyclonal antibody that binds to corresponding antigen epitopes. This indicates that avidity measurement is more suitable than affinity for measuring antigen affinity of polyclonal antibodies in blood. The conventional avidity measurement is based on the removability of antibodies bound to antigen epitopes by washing with chaotropic agents such as urea for a short time under nonequilibrium conditions [7,8] and has provided a large number of evidence regarding the avidity maturation of IgG to SARS-CoV-2 antigen, and their ability to protect against infections induced by several steps of vaccination much higher than natural infection [7,8,9,10,11,12]. In addition, the importance of high avidity index for protective immunity has been established for many infections with viruses and other microbes [8].

To measure avidity index, enzyme-linked immunosorbent assay (ELISA) is conventionally used in the absence and presence of various concentrations (4–8 M) of chaotropic agents over various short-time treatments (3–10 min) in the washing buffer to remove low avidity IgG [7,8,9,10,11,12]. This method is called chaotropic avidity measurement in this paper. However, there are limitations to the measurement of avidity index in the assay system described above. Treatment of urea (which gently destructs the three-dimensional structure of proteins and modifies the conformation of antibodies and antigens in the reaction system) does not show the direct strength of binding interaction between native antigen and antigen-specific antibodies. Furthermore, in the above measurement system, the avidity index values under nonequilibrium conditions vary depending on the antibody concentration in the reaction mixture even for the same urea concentration and it is difficult to evaluate or compare the binding interactions reported by various research groups.

To overcome such limitation and to accurately determine the protective immunity of the antibodies, we used a new parameter in this study to measure the antigen binding avidity of the variable region of the IgG antibodies. The binding avidity assay was first used in the field of food allergy diagnosis by the densely carboxylated protein (DCP) microarray [13] by us as a method to increase the diagnostic performance of antigen-specific IgE levels in the deferential diagnosis of food allergy and anaphylaxis, even when it is difficult to diagnose by food challenge testing, and its potential usefulness has been proven [13,14,15,16]. The DCP microarray uses a new surface chemistry of carboxylated arm technology to cover the surface of a glass slide or diamond-like carbon-coated chip to immobilize proteins and DNA at very high density of 0.94–7.82 × 10^9^ molecules mm^−2^, as described in detail previously [17]. The binding avidity assay was also found to be useful in the evaluation of IgG antibody’s ability to protect against infection in the field of infectious diseases.

Accumulating evidence suggests that the infectious pathogenic antigen-capturing potency of antibodies, which reflects their ability to protect against infection, is not homogeneous, and that this is true even for the same subtype of antigen-specific antibodies. This means that the traditional methods used for measuring antibody titer, i.e., concentration of antibody with a secondary antibody that recognizes the constant region of the antibody (see illustration in Figure 1A), cannot evaluate the overall protective immunity of the antibodies, because such assays do not measure the strength of the binding reaction to the antigen carried by the variable region of the antibody. As shown in Figure 1B, the RBD antigen binding avidity of antibodies is quantitatively measured by the competitive binding inhibition of the antibodies between the immobilized SARS-CoV-2 S1 protein containing the RBD on the DCP microarray and the serially diluted soluble RBD protein in the test sample, and the IC50 (nM) value is determined after washing to remove unbound anti-RBD antibodies. IC50, the half-maximal inhibition concentration, represents the concentration of RBD protein required for 50% binding inhibition. Using this method, the antigen binding avidity was defined as the level of 1/IC50 (nM). We then used antigen binding avidity antibody titer (ABAT) as an index of protective immunity, which is expressed by the antibody quantity multiplied by the quality (antigen binding avidity, 1/IC50). Low IC50 values represent high binding avidity IgG while high IC50 values represent low binding avidity IgG (Figure 1B).

Increases in the maturation of the antigen binding avidity of antipathogen-specific IgG to the target epitopes are triggered by repeated vaccinations or by infections. This is due to B cell maturation over time that occurs in the germinal center, leading to maturation of antibodies and enhancing their binding avidity for their cognate antigen [11].

Here, we describe our new method to evaluate the maturation of the protective immunity of anti-SARS-CoV-2-specific antibodies by measuring the RBD binding avidities and their ABATs. Our results showed that the increase in infection protection potency of anti-SARS-CoV-2 specific antibodies following SARS-CoV-2 infection or repeated COVID-19 messenger RNA (mRNA) vaccination, as expressed by ABAT values, can be classified into four groups: anti-SARS-CoV-2 specific antibodies below the limit of detection and three anti-SARS-CoV-2 specific antibodies with different levels of maturity of infection defense capacity; (i) the group of ABAT antibodies with an initial RBD binding avidity, (ii) the group of ABAT antibodies with low RBD binding avidity, and (iii) the group of ABAT antibodies with high RBD binding avidity. These ABAT antibodies correlated closely to the antiviral neutralizing activities and also with the clinical severity of hospitalized COVID-19 patients.

## 2. Materials and Methods

### 2.1. Study Population and Plasma Samples

Serum samples were obtained from 163 healthy vaccinated volunteers (hospital staff of National Hospital Organization Mie National Hospital) aged 21–71 years. All subjects had received the first dose of the BioNTech/Pfizer mRNA vaccine (COMIRNATY^®^/BNT162B2) in March 2021, followed by second vaccination three weeks later, third vaccination at 38–40 weeks, and fourth vaccination at 71–74 weeks after the first vaccination. Blood samples were collected 3–4 weeks after each vaccination (Table 1). We also included in this study 339 unvaccinated COVID-19 patients, aged 14 to 98 years, who had been admitted to three separate hospitals for acute SARS-CoV-2 infection, as confirmed by examination of pharyngeal swabs by polymerase chain reaction (PCR) test, between the period of April 2020 to August 2021 prior to the Omicron variant epidemic in two distant regions of Tokyo and Okinawa in Japan [18,19]. Based on the admission interview, these patients were determined to be not previously infected with SARS-CoV-2 and naïve to specific treatment for COVID-19, including remdesivir or favipiravir [20,21]. Disease severity was determined based on the COVID-19 severity classification developed by the Ministry of Health, Labor, and Welfare (MHLW) of Japan [22]. Sera collected from the COVID-19 patients and vaccinated individuals were stored at −20 °C until analysis.

### 2.2. Ethics Statement

The study protocol, informed consent, and data collection of the healthy vaccinated volunteers and COVID-19 patients were reviewed and approved by the Institutional Review Board of the Tokushima University Hospital (#4067 and #4068-1), and a written informed consent was obtained from each participant.

### 2.3. Antigens and Reagents

HEK293 expressing recombinant His Tag SARS-CoV-2 S1 of the original Wuhan strain (Accession# QHD43416.1, Val16–Arg685) and S protein RBD (Arg319–Phe541) were purchased from ACROBiosystems (Newark, DE, USA). Antihuman IgG-Fc fragment antibody (goat polyclonal) was from Bethyl Laboratories, Inc., Montgomery, TX, USA).

### 2.4. Anti-Spike S1 IgG Quantification

Serum titers of anti-spike S1-specific IgG were measured by ELISA using 96-well plates [23] or DCP microarrays, as described in detail previously [13,14,15,16,17]. RBD binding avidity was also measured using DCP microarrays. A 96-well plate (Nunc, Naperville, IL, USA) coated with S1 protein (0.1 μg/well) in ELISA Coating Buffer (Bethyl Laboratories) was used for ELISA assay. Anti-spike S1-specific IgG titers were also measured by fluorescence immunoassay on the DCP microarray. Briefly, S1 antigen (4.8 ng) in phosphate-buffered saline (PBS) spotted on the DCP microarray was incubated for 1 h with 8 μL of 1:20 to 1:6000 diluted serum, washed three times with TTBS (50 mM Tris-HCl, pH 7.5, containing 150 mM NaCl and 0.05% Tween 20), rinsed with deionized water, and then reacted with a HiLyte Fluor 555 (Dojindo Molecular Technologies, Inc., Kumamoto, Japan)-labeled secondary goat polyclonal antibody against human IgG-Fc fragment at a concentration of 30 ng/mL in dilution buffer (0.5% bovine serum albumin, 0.05% Tween 20 and 0.3 M KCl in PBS) for 1 h. The resulting images were acquired by scanning the microarrays with an InnoScan 710 (Innopsys, Carbonne, France), and converted to numerical fluorescence intensity. Using standard concentrations of purified human IgG (Fujifilm Wako Pure Chemicals, #143-09501, Osaka, Japan), the amounts of IgG bound to the microarrays were calculated and expressed as binding units (BUs). The detection limit is 200.0 BUg/mL under the present assay conditions. We reported previously the intra- and inter-assay coefficients of variation for the DCP microarray assay as follows: inter-assay: 5.70% to 13.5%, within-slide: 7.70% to 25.2%, and batch-to-batch: 2.7% to 24.4% [17]. As shown in Appendix A, the S1-specific IgG titers analyzed by the DCP microarray correlated strongly with those determined by ELISA (*r* = 0.8964).

### 2.5. RBD Binding Avidity Measurement of Anti-Spike S1 IgG Antibodies

The level of anti-S1-IgG in each serum sample bound on the DCP microarray was adjusted at fluorescence intensities between 5000 and 60,000 by dilution with PBS containing 0.3 M KCl and 0.05% Tween 20 before analysis of RBD binding avidity. RBD protein at 0, 2, 20, 200 ng as a binding competitor of anti-spike S1 IgG antibodies was then added to the reaction mixture (16 μL) of serum and then preincubated at 37 °C for 30 min. After the preliminary reaction, each reaction mixture (8 μL) was applied to an S1-immobilized DCP microarray and competitive binding assay was conducted under equilibrium conditions at 37 °C for 1 h. After washing three times with TTBS, rinsing with deionized water, followed by spin-down drying, the antibodies on the DCP microarray were detected by reaction with a HiLyte Fluor 555-labeled secondary antibody against human IgG-Fc fragment at a concentration of 300 ng/mL at 37 °C for 1 h. After the same washing and drying as described above, we measured the fluorescence intensity using a scanner, followed by calculation of IC50.

### 2.6. Neutralizing Activity of Antibodies against Pseudovirus

Lenti-X 293T cells and human ACE2 293T cells were cultured in DMEM (high glucose: 4.5 g/L), containing 4 mM L-glutamine and 3.7 g/L sodium bicarbonate, 10% fetal bovine serum, 100 units penicillin G sodium, 100 μg/mL streptomycin sulfate, and 1 mM sodium pyruvate. Lenti-X™ SARS-CoV-2 Packaging Single Shots (D614G Spike, Truncated) (Takara Bio Inc., Shiga, Japan) were prepared with Lenti-X 293T cells, according to the protocol provided by the manufacturer. The concentrations of pseudoviruses incorporating SARS-CoV-2 (mutation of D614G, deletion of the final 19 amino acids of the C-terminus) in the pLVXS-ZsGreen1-Puro Vector were measured using Lenti-X GoStix Plus (Takara Bio Inc.).

To measure the pseudovirus-neutralization activity of anti-S1-IgG antibodies, human ACE2 293T cells were seeded onto 96-well plates at 4 × 10^4^ cells per well and infected with pseudovirus after 24 h incubation at 37 °C. Pseudovirus (100 ng/well) and serum (dilution: 2^5^-fold to 2^10^-fold) in the culture medium were incubated for 1 h at room temperature to infect the above human ACE2 293T cells in the presence of 6 μg/mL of polybrene followed by 72 h incubation. Before measurement, blood samples were classified into three categories of binding avidity (1/IC50) maturation according to the correlation between anti-SARS-CoV-2 IgG antibodies and ABAT values, described in detail later: initial binding avidity (0.001–0.004 nM), low binding avidity (0.005–0.009 nM), and high binding avidity (0.053–0.077 nM). After culture, the medium was aspirated, and the cells were fixed with 4% paraformaldehyde for 30 min at room temperature. After washing with PBS, the ZsGreen1-expressing infected cells were photographed with a fluorescence microscope (BZ-X700, KEYENCE Corp., Osaka, Japan) and counted with the analysis application software provided with the microscope. The number of infected cells in the absence of serum was set as 100%, and the number of infected cells after treatment with pseudovirus and neutralizing serum in each assay group was counted.

### 2.7. Statistical Analysis

Receiver operating characteristic (ROC) curve and area under the curve (AUC) statistical analyses were performed using JMP software, version 14.2.0 (SAS Institute Inc., Cary, NC, USA). Correlation analyses were performed using Excel software (Microsoft Corp, Redmond, WA, USA). *r* ≥ 0.7 was considered high correlation. Data were expressed as mean ± SEM.

## 3. Results

### 3.1. Changes in Anti-SARS-CoV-2 Spike S1 Specific IgG Antibody Titer and Its RBD Binding Avidity in Vaccinated Subjects and SARS-CoV-2-Infected Patients

Data of anti-SARS-CoV-2 spike S1 IgG antibody titer and its RBD binding avidity in serum of 163 healthy vaccinated subjects who received one to four BioNTech/Pfizer mRNA vaccinations are shown on the right-side panels of Figure 2A,B. The antibody titers and RBD binding avidities before vaccination were extremely low or below the detection limits. After 3 weeks of the first vaccination, antibody titers increased in almost all subjects, and then reached near-plateau level after 3 weeks of the second vaccination, with the exception of a very small number of subjects. However, the antibody titers then decreased to about 34.5% of the plateau levels after 6–8 months of the second vaccination, just before the third vaccination. The levels increased again to the plateau levels at 3 weeks after the third vaccination. The change in antibody titers after 6–8 months of the third vaccination was different from that after the second, with the antibody titers remaining at high levels at about 83.3% of the plateau levels even at 6–8 months after the third vaccination. Interestingly, no significant changes were observed in antibody titers after the fourth vaccination, with a slight increase in the plateau levels.

Figure 2B shows changes in RBD binding avidity (1/IC50) of anti-SARS-CoV-2 spike S1 IgG antibody. RBD binding avidity levels before vaccination were below the detection limit in all subjects but increased broadly in almost all subjects at 3 weeks after the first vaccination. In some cases, the increased levels were close to the plateau levels of the mature RBD binding avidity found after the third vaccination. More than half of the other subjects showed lower binding avidity levels. The increased levels of RBD binding avidity were about 90.2% of the plateau levels after the second and 100% after the third vaccination, and no further significant increase in the levels were observed after the third vaccination. In addition, once high binding avidity was achieved, it was maintained for at least 6–8 months regardless of subsequent decreases in antibody levels.

Figure 2A,B (left panels) show SARS-CoV-2-specific IgG antibody titers and their RBD binding avidities in sera of 339 SARS-CoV-2-infected unvaccinated patients, measured on the day of admission and at 2–3 weeks after admission or at discharge. On admission, 76% of patients had antibody titers below the assay detection limit of 200 BUg/mL and 81% of patients had RBD binding avidity below the detection limit of 0.0002 nM. Although antibody titers and their RBD binding avidities increased during the hospitalization period in a manner similar to the first BioNTech/Pfizer mRNA vaccination in the healthy volunteers, the antibody titers of 32% patients and their binding avidities in 36% of the patients were still below the detection limits. This finding indicates that the efficiency of acquiring protective immunity in patients against SARS-CoV-2 infection is inferior to that of mRNA vaccination. A small number of hospitalized patients, approximately 3% for antibody titers and 6% for RBD binding avidity values, showed high values at the time of admission, nearly the plateau levels of the mRNA vaccination, despite no previous infection or vaccination history, and these values represented only mild increases of 7% and 19%, respectively, during the hospitalization. Most of these patients were discharged after 2–3 weeks of hospitalization, but their antibody titers and RBD binding avidity at discharge were much lower than those achieved in subjects who received two or three vaccinations.

### 3.2. Three-Stage Maturation of RBD Binding Avidity of Anti-SARS-CoV-2 Antibodies Provides Protection against Infection

Figure 3A shows the broad distribution of antibody titers in the abscissa and their RBD binding avidities in the ordinate after the first vaccination. The data show three regression lines with different maturation levels of antibody titers and their RBD binding avidities. Among the three lines, the group with the baseline binding avidity ABAT is designated “initial binding avidity ABAT”, the group with the high binding avidity ABAT is labeled “high binding avidity ABAT”, and the group with binding avidity ABAT somewhat between the two groups is labeled as “low binding avidity ABAT”. During the progression of antibody maturation, the “initial binding avidity ABAT” disappeared after the second vaccination (Figure 3B), then the “low binding avidity ABAT” disappeared after the third vaccination (Figure 3C), and the “high binding avidity ABAT” was only detected after the third and fourth vaccinations (Figure 3C,D).

We also assessed the infection-protective immunity of the three ABAT antibodies with different RBD binding avidity maturation by virus-neutralizing activity using pseudovirus (Figure 4). The “high binding avidity” group with 1/IC50 ≥ 0.02 nM showed dose-dependent potent virus-neutralizing capacity with reduction in the percentage of infected cells to about 1% with serum dilution fold at 32. Antibodies with the “low binding avidity” group with 0.02 nM > 1/IC50 > 0.0005 nM and the “initial binding avidity” group with 1/IC50 ≤ 0.0005 nM showed lower neutralizing capacities, with the number of infected cells as high as roughly 20% and 90%, respectively, with serum dilution fold at 32. The results showed a clear difference in virus-neutralization potencies with different maturation of RBD binding avidities (Figure 4B). The correlation coefficient *r* value between the RBD binding avidity (1/IC50) values of anti-SARS-CoV-2 antibodies and their virus-neutralizing activities analyzed by using pseudovirus indicated a relatively high correlation (*r* = 0.811) (Appendix A).

### 3.3. Receiver Operating Characteristic (ROC) Analysis to Determine Cutoff Values of Anti-SARS-CoV-2 Antibodies of Vaccinated and Nonhospitalized Subjects from Nonvaccinated and Hospitalized COVID-19 Patients

In the next step of our analysis, we used ROC analysis to assess the protective efficacy of the induced anti-SARS-CoV-2 antibodies in the normal vaccinated subjects and COVID-19 patients. The aim of this step was to determine the cutoff values of antibody titer, RBD binding avidity, and ABAT to predict protection against hospitalization (Figure 5).

ROC analysis was performed on anti-SARS-CoV-2 antibodies of COVID-19 patients at admission and those of vaccinated normal subjects at 3–4 weeks after the third vaccination. ROC analysis for anti-spike S1 antibody titer (Figure 5A) showed area under the curve (AUC) value of 0.975, with sensitivity of 96% and specificity of 100%, and optimal cutoff value of 2121 BUg/mL. ROC analysis for RBD binding avidity (IC50) (Figure 5B) showed AUC value of 0.987, with sensitivity of 96% and specificity of 99%, and optimal cutoff value of 40.6 (nM). Furthermore, ROC analysis for ABAT (Figure 5C) showed that these values were 0.980, 97%, 100%, and 74.4 ((BUg/mL)/(IC50 nM)), respectively.

These cutoff values of anti-SARS-CoV-2 S1-specific antibodies for the protective immunity that can protect against hospitalization were then plotted to show the change in RBD binding avidity maturation with increasing number of vaccinations (from the first to the third), to elucidate the number of vaccinations required to avoid hospitalization (Figure 6A–D). The abscissa in Figure 6 shows the natural logarithm of the anti-SARS-CoV-2 S1-specific antibodies (BUg/mL) while the ordinate shows that of ABAT ((BUg/mL)/IC50 (nM)). Each plot includes the regression lines of different RBD binding avidity maturation and their 95% confidence prediction intervals after the first (Figure 6A), second (Figure 6B), and third (Figure 6C) vaccination. Figure 6D shows the distribution areas for the three different RBD binding avidity maturation antibodies with cutoff values for the prediction of protection against hospitalization: initial binding avidity ABAT, low binding avidity ABAT, and high binding avidity ABAT. These data suggest that mature high binding avidity ABAT after the third vaccination resulted in immunity that could safeguard against hospitalization.

### 3.4. Analysis of Relationship between ABAT Levels on Admission and Severity of SARS-CoV-2 Infection Outcome after 2–3 Weeks of Hospitalization

Anti-SARS-CoV-2-specific antibody titers and their RBD binding avidities of the majority (approximately 66%), but not all, the hospitalized patients increased after 2–3 weeks admission (Figure 2), and their data were replotted to assess the change in defense immunity during hospitalization (Appendix A).

The antibody titers (65%) and ABATs (58%) of the majority of patients at 2–3 weeks after admission or at discharge distributed in the areas of initial and low binding avidity ABAT, with only a small group of patients in the high antibody titer (3%) and the high binding avidity ABAT (6%) (Figure 2). The rest of the patients showed no increase in the antibody titers (32%) and their RBD binding avidities (36%), as shown in the heavily overlapping intersection of the abscissa and ordinate in the left-side corner of Appendix A. These results suggest that the protective immunity of hospitalized patients was less effective compared with the efficacy of a single-dose vaccination with the mRNA COVID-19 vaccine.

Figure 7 summarizes the relationship between ABAT levels of anti-SARS-CoV-2-specific antibodies at admission and outcome of infection severity after 2–3 weeks of hospitalization or at discharge. The three ABAT categories; initial, low, and high binding avidity ABATs, were associated with different severity of infection after hospitalization. Most patients with high binding avidity ABAT showed mild symptoms (77.8%) and none showed severe illness. On the other hand, 77.2% and 56.2% of the patients with low and initial binding avidity ABAT showed severe or moderate infection, respectively, and only 22.8% and 43.8%, respectively, showed mild symptoms.

## 4. Discussion

It has become clear in recent years that even the same concentration of antibodies has diverse binding potencies to antigens. The defense capability of IgG against infection is also diverse, and the antibody titer, which is estimated by using a secondary antibody that recognizes the constant region of IgG molecules and measures the concentration of specific IgG molecules, is insufficient for the full assessment of the diverse defense capability of the IgG. The ABAT parameter evaluated in this study assesses more accurately and directly the ability to capture antigens as an antibody function using the antigen binding avidity parameter. As shown in Appendix A, the RBD binding avidity (1/IC50 nM) correlated highly (*r* = 0.811) with the virus-neutralizing activity of the antibodies using pseudovirus. The RBD binding avidity assay system uses the DCP microarray [13,14,15,16,17], which has been certified in Japan for quantitative antigen-specific IgE analysis, based on the high-quantification data and reproducibility.

Assessment of the defense capability of anti-COVID-19 antibodies by ABAT has the following advantages compared to the pseudovirus-neutralization assay using the cell culture system. (i) Short processing time: The results of the conventional method of virus-neutralizing activity using pseudovirus and cultured cells become available after a minimum period of 3–5 days from the start of cell culture, whereas the results of the ABAT assay by DCP microarray are ready within about 5 h. (ii) Several hundred specimens can be processed simultaneously and rapidly in the ABAT assay, whereas this is practically difficult in the virus-neutralizing activity using the cell culture system. (iii) Using the DCP microarray, the ABAT assay provides a comparable unit of antigen binding avidity and quantitative measure (1/IC50 RBD antigen concentration), instead of the 50% viral neutralization activity represented by the dilution fold of the specimen. In other words, the ABAT assay allows comparison of different specimens.

Since the protection potential of anti-COVID-19 antibodies could be quantitatively expressed by the RBD binding avidity (1/IC50 nM) and the ABAT value, it is possible to classify the antibodies into three categories based on differences in the binding avidity maturation (see Figure 4). A high RBD binding avidity and ABAT values of antibodies with the highest protection capacity against infection could be set as goal target values for vaccination and protective antibodies in the infected patients. By using these data, it could be possible to estimate the need for additional vaccination(s) and assess clinical outcome in patients with severe infection. In patients with high RBD binding avidity on admission, none was severely ill and 77.8% were mildly ill at 2–3 weeks of hospitalization or at discharge (Figure 7). In contrast, 77.2% and 56.2% of patients with low and initial RBD binding avidity on admission had moderate and severe illness, respectively, at 2–3 weeks of hospitalization or at discharge. Thus, the RBD binding avidity and ABAT well reflect the severity of illness in hospitalized patients. A similar relationship between avidity index analyzed by chaotropic avidity and severity of SARS-CoV-2 patients was reported by Georg Bauer [8].

Maturation of the RBD binding avidity of antibodies induced by SARS-CoV-2 infection for 2–3 weeks was weaker than those at 3–4 weeks after the first vaccination with BioNTech/Pfizer mRNA, and approximately 32–36% of the patients showed neither an increase in antibody titer nor RBD binding avidity. These results suggest that the immunogenicity of the live SARS-CoV-2 virus is lower than the mRNA vaccine. Most patients showed low and initial RBD binding avidity, and high RBD binding avidity was noted in only 6% of these patients. Based on the above findings in the infected patients, we recommend mRNA vaccination of the patients before or after discharge to achieve antiviral antibody ABAT levels that could prevent rehospitalization.

As shown in Figure 2B, the speed of maturation of the RBD binding avidity, which reflects the ability to protect against infection, varied greatly among individuals. Although most of the vaccinated individuals (90.2%) achieved 1/IC50 of 0.02 nM or higher at 3–4 weeks after the second vaccination, some (9.8%) showed slow maturation that did not reach 0.02 nM. However, even among vaccinated subjects who showed slow maturation, all achieved 1/IC50 of 0.02 nM or higher at 6–8 months after vaccination without booster vaccination. These findings suggest that the second and subsequent vaccinations can be applied after a long interval of 6–8 months, anticipating the maturation of the pre-existing immunity. This result supports the concept of “longer vaccination interval improves SARS-CoV-2 neutralization” reported recently [24].

Our study has certain limitations. First, we could not follow up changes in antibody titer, RBD binding avidity, and ABAT values after 2–3 weeks of hospitalization; thus, future long-term follow-up studies are planned. Second, the number of study subjects needs to be increased in the future to determine the effect of the underlying disease on the analyzed parameters. Third, the present study was conducted before the emergence of SARS-CoV-2 Omicron variants, but it is necessary to investigate the protective potential and cross-reactivity of SARS-CoV-2 wild-type antibodies against antibodies generated during infection with SARS-CoV-2 Omicron variants, using antibody titers, RBD binding avidity, and ABAT values.

In conclusion, we described in this report the development of a new method to measure the antigen binding avidity/affinity of antibodies involved in infection defense capacity. Conventional methods of determination of antibody titer, which measure antigen-specific antibody levels, do not provide the diverse antigen binding potencies, which are well found among antibodies. In this study, the antigen binding capacity expressed by the variable region of the antibody was measured by the competitive binding inhibition (IC50) between the antigen immobilized on the DCP microarray and the soluble antigen added to the test sample. Thus, the quantity × quality (binding avidity) of antibodies, ABAT, was successfully used to quantify the ability of the antibody to protect against infection. The values of RBD binding avidity correlated strongly and significantly with cell-based virus-neutralizing activity. In addition, the new method allows the monitoring the maturation process of RBD binding avidity, as well as the classification of antibodies into three categories based on the differences in antibody maturation: initial, low, and high ABAT. Furthermore, once high RBD binding avidity was achieved, it was maintained for at least 6–8 months regardless of the subsequent decrease in antibody levels. These characteristics may prove useful in formulating effective vaccination programs.

## Figures and Tables

**Figure 1 viruses-15-01662-f001:**
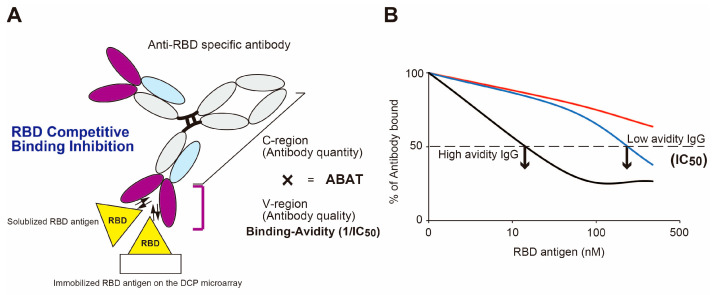
RBD binding avidity assay. (**A**) Concept of “antigen binding avidity antibody titer (ABAT)” expressed by the antibody quantity multiplied by the antigen binding avidity, 1/IC50. (**B**) Schematic presentation of the method of antibody-mediated competitive antigen binding inhibition between the immobilized S1-RBD protein on the DCP microarray and solubilized RBD antigen for determination of the RBD IC50 (nM). Arrows indicate the IC50 concentrations.

**Figure 2 viruses-15-01662-f002:**
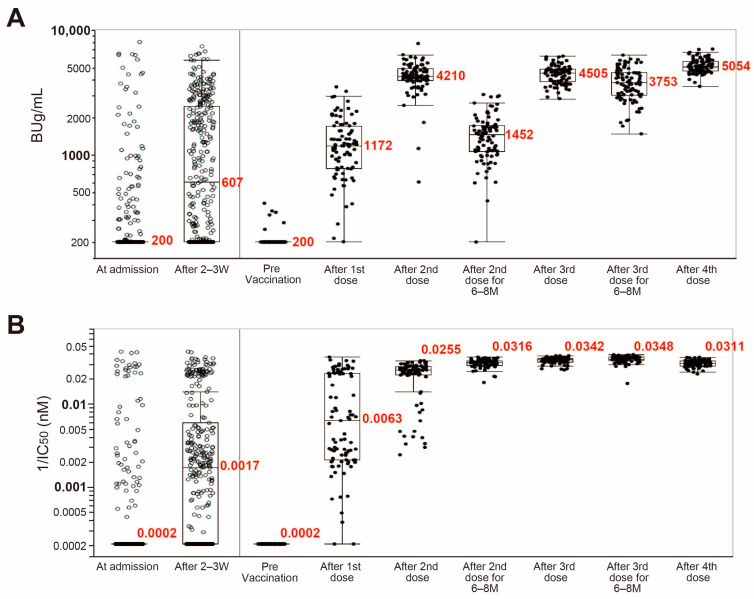
Changes in anti-spike S1 IgG antibody titer and its RBD binding avidity in vaccinated subjects following 2–4 boost immunizations and in SARS-CoV-2-infected patients. (**A**) Antibody titer (BUg/mL). (**B**) RBD binding avidity (1/IC50, nM). Blood samples of healthy vaccinees (*n* = 163, closed circles) were collected 3–4 weeks after each vaccination at the indicated times in Table 1 and those of unvaccinated COVID-19 patients (*n* = 339, open circles) on admission and 2–3 weeks after admission or at discharge. The median (interquartile range) values are in red. M: month.

**Figure 3 viruses-15-01662-f003:**
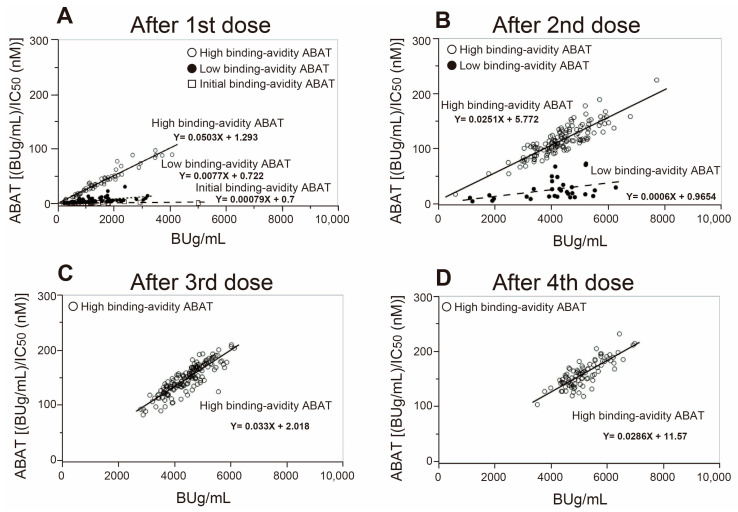
Correlation between anti-SARS-CoV-2 IgG titers and their ABAT values at 3–4 weeks after 1st to 4th vaccinations showing a maturation of the antigen binding avidity. (**A**) After 1st vaccine, (**B**) after 2nd vaccine, (**C**) after 3rd vaccine, and (**D**) after 4th vaccine (*n* = 163). Note the differences in the correlation regression lines for different antigen binding avidities.

**Figure 4 viruses-15-01662-f004:**
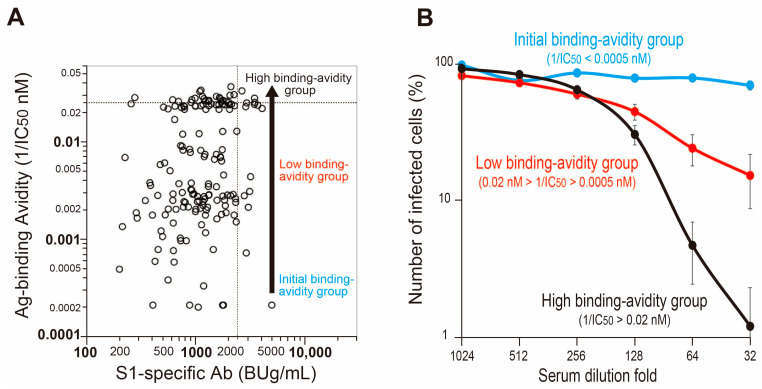
Binding avidity and virus-neutralizing maturation of antiviral S1-specific antibodies by vaccination. (**A**) Three different RBD binding avidities (ordinate) with similar levels of antiviral S1-specific antibodies (abscissa) induced by 1st vaccination. Arrow indicates direction of binding avidity maturation. Dotted lines on the ordinate and abscissa axes show each cutoff value from Figure 5. Ag: antigen; Ab: antibody. (**B**) Serum dilution-fold-dependent virus-neutralizing activities of three S1-specific antiviral antibodies with different binding avidity maturation. Data are mean ± SEM (*n* = 8–27). The range of IC50 values for the three S1-specific antiviral antibodies with different binding avidity maturation is shown in parentheses.

**Figure 5 viruses-15-01662-f005:**
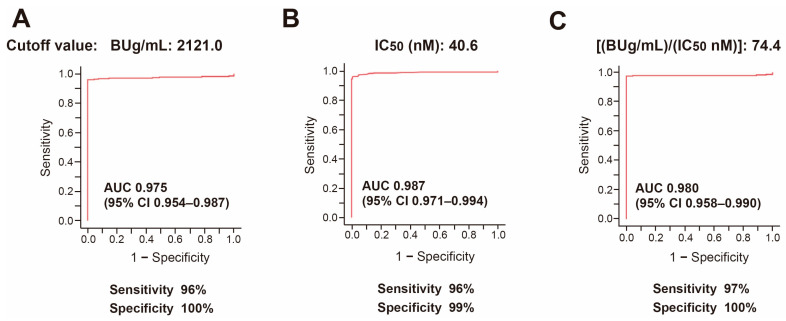
ROC analysis to determine the cutoff values of serum anti-SARS-CoV-2 antibody level, IC50, and ABAT in vaccinated normal subjects and COVID-19 patients. (**A**) Anti-SARS-CoV-2 antibody level (BUg/mL), (**B**) IC50 (nM), (**C**) ABAT ((BUg/mL)/(IC50 nM)). AUC: area under the curve; CI: confidence interval.

**Figure 6 viruses-15-01662-f006:**
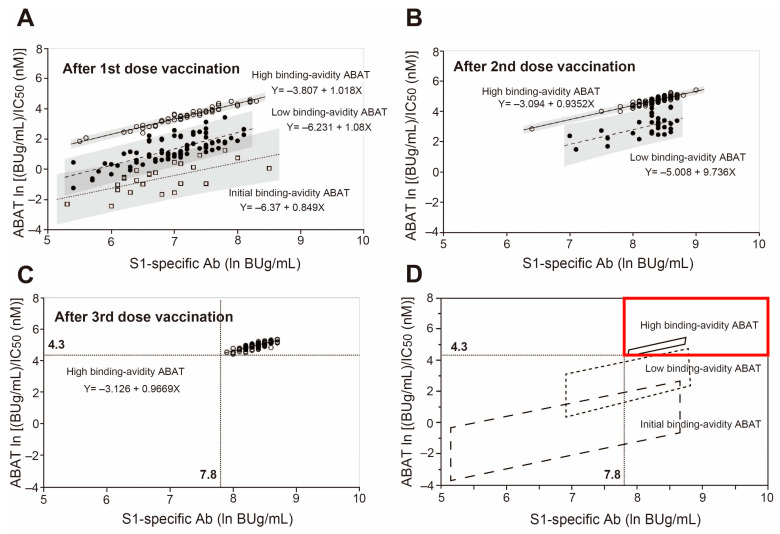
Maturation of binding avidity with repeated vaccinations and determination of cutoff values of COVID-19 antibodies to avoid hospitalization. Correlation regression lines with 95% CI areas between anti-SARS-CoV-2 IgG titers (In BUg/mL) and ABAT In ((BUg/mL)/IC50 (nM)) after 1st (**A**), 2nd (**B**), and 3rd (**C**) vaccinations, and schematic diagram (**D**) showing the three areas of ABATs with different binding avidity maturation: high binding avidity ABAT area from (**C**), low binding avidity ABAT area from (**B**), and initial binding avidity ABAT area from (**A**). Dotted lines in (**C**,**D**) show cutoff values of S1-specific antibody titer and ABAT for predicting avoidance of hospitalization. Regions exceeding the above cutoff values are indicated by red box. Hollow circles, antibodies with high binding-avidity ABAT; Black solid circles, antibodies with low binding-avidity ABAT; Squares, antibodies with initial binding-avidity ABAT; Ab, antibody.

**Figure 7 viruses-15-01662-f007:**
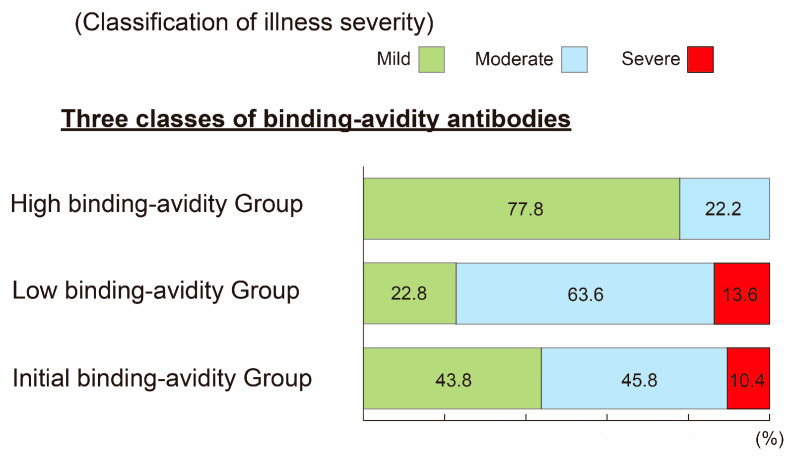
Correlation between anti-SARS-CoV-2 IgG antibodies with three different binding avidity maturations on admission and classification of severity of illness at 2–3 weeks after admission or at discharge. RBD binding avidity maturation of anti-SARS-CoV-2 IgG antibodies was measured and classified into three groups on admission, high binding avidity group (*n* = 9), low binding avidity group (*n* = 23), and initial binding avidity group (*n* = 307) and classification of severity of illness at 2–3 weeks after admission or at discharge.

**Table 1 viruses-15-01662-t001:** Details of serum collection from patients and healthy vaccinated subjects.

		Gender			Time of Serum Collection
	*n*	M (*n*)	F (*n*)	Age (Years)	(Weeks after Initial Vaccination)
Healthy subjects	163	39	124	41 (21–71)	
First vaccination					March 2021 (3 weeks)
Second vaccination					April 2021 (6–7 weeks)
Third vaccination					December 2021 (41–44 weeks)
Fourth vaccination					September 2022 (74–77 weeks)
COVID-19 patients	339	204	135	51 (14–98)	on admission (April 2020 to August 2021)
	339	204	135	51 (14–98)	on 2–3 weeks after admission or at discharge (May 2020 to September 2021)

## Data Availability

The data presented in this study are available in the article and the Appendix A here.

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
