# Peer review of "Clinical Utility of SARS-CoV-2 Antibody Titer Multiplied by Binding Avidity of Receptor-Binding Domain (RBD) in Monitoring Protective Immunity and Clinical Severity"

_viruses, 2023, doi:10.3390/v15081662_

Round 1

Reviewer 1 Report

Takahashi et al. present a well-designed, perfectly controlled and excellently performed study on the functional quality of IgG directed towards RBD of SARS-CoV-2 after variable steps of vaccination or after natural infection. They utilize a novel assay based on subsequent competition/inhibition reactions. First, a serum dilution (adjusted for suitable concentration range of specific IgG) is allowed to react with various concentrations of soluble RBD for a relatively short time (30 min), and then this mixture is added to reaction wells in which spike protein of SARS-CoV-2 at defined concentration is fixed to the surface. The second incubation is carried out for an additional hour. In this assay, IgG with low binding strength requires relatively high concentrations of soluble RBD during the first incubation for competition, which is relatively low. IgG with very high binding strength only requires very low concentrations of competing RBD during the first incubation and very efficient competition can be achieved. This system works in a concentration-dependent mode and leads to highly significant values of binding strength, calculated as 1/ IC50.
The authors used this impressive assay system to demonstrate the increase in binding strength after increasing numbers of vaccination steps. They show that binding strength after first vaccination is more variable between individuals than after second vaccination (where binding strength is also higher) and they demonstrate that third and fourth vaccination leads to further increase of binding strength with extremely low variability. The authors show that individuals with low binding strength of IgG towards RBD have relatively higher risk of more severe disease than individuals with IgG of higher binding strength. These data and the specific assay that was used to generate these data are of high significance for our understanding of humoral immunity after vaccination towards SARS-CoV-2 and for the evaluation of protective immunity. Therefore, this fine study should be presented to the scientific community.

To further increase the significance and quality of this work, the authors are advised to include the following additional information, derived from the previous work of other authors, into Introduction and/or Discussion of this manuscript:

1) The author should mention that classical avidity determination, using chaotropic agents after an initial binding of IgG to SARS-CoV-2 antigens has shown that i) several steps of vaccination lead to much higher avidity than natural infection with SARS-CoV-2. This finding is in contrast to avidity maturation after other viral infection and seems to be unique for coronaviruses (references: Struck F et al., Vaccination versus infection with SARS-CoV-2: Establishment of a high avidity IgG response versus incomplete avidity maturation. J Med Virol, 93: 6765-6777, 2021; doi: 10.1002/jmv.27270, reviewed in Bauer G. High avidity of vaccine-induced immunoglobulin G (IgG) against SARS-CoV-2: potential relevance for protective humoral immunity. Explor Immunol. . 2022;2:133–156. DOI: https://doi.org/10.37349/ei.2022.00040).

2) The authors should mention that their clear finding of lower clinical severity of infection in patients with IgG of high binding strength correlates very well with the findings by Moura et al., Ravichandran et al., Tang et al. and others, as reviewed in Bauer G. High avidity of vaccine-induced immunoglobulin G (IgG) against SARS-CoV-2: potential relevance for protective humoral immunity. Explor Immunol. . 2022;2:133–156. DOI:
https://doi.org/10.37349/ei.2022.00040).

3) The authors should mention that the importance of high avidity for protective immunity has been established for many infections with viruses and other microbes (reviewed in Bauer G. High avidity of vaccine-induced immunoglobulin G (IgG) against SARS-CoV-2: potential relevance for protective humoral immunity. Explor Immunol. . 2022;2:133–156. DOI: https://doi.org/10.37349/ei.2022.00040) .
This is in perfect agreement with the author`s statement at the beginning of “Discussion”.

4) The authors should clearly distinguish in their introduction between “avidity” and “affinity” of IgG. Avidity defines the binding strength between IgG and its epitope and is usually measured by first allowing a sufficiently long reaction between test antigen and serum, followed by a step that determines to which degree IgG can be removed by chaotropic agents. As the basis for avidity is dependent on the affinity of IgG, avidity determination gives a good indication of relevant binding strength. In contrast, affinity is defined as the sum between forward and reverse reaction between epitopes and IgG. However, this can not easily be measured in heterogenous mixtures of IgG towards different epitopes and also being at different levels of affinity maturation. Definitions of avidity and affinity and their measurements can be found in the following publications: Hedman K, et al. Avidity of IgG in serodiagnosis of infectious diseases. Rev Med Microbiol. 1993;4:123–9; . Nurmi V et al., Comparison of approaches for IgG avidity calculation and a new highly sensitive and specific method with broad dynamic range. Int J Infect Dis. 2021;110:479–87; Bauer G. High avidity of vaccine-induced immunoglobulin G (IgG) against SARS-CoV-2: potential relevance for protective humoral immunity. Explor Immunol. . 2022;2:133–156. DOI: https://doi.org/10.37349/ei.2022.00040) .

5) Based on the considerations from 4), it seems that the assay system used in the study by Takahashi et al is not based primarily on avidity of IgG, but is controlled by affinity of IgG. I can not figure out in which way avidity would determine the concentration of competing RBD (added for the preincubation step) required for a certain degree of competition, whereas it is very easy to understand that IgG of high affinity would react very fast, even with lower concentrations of competing RBD, and in this way would not be available for subsequent binding to spike protein fixed to the bottom of the test tube. It might be helpful if the authors tried to modifiy Figure 1 (which is not very illustrative at present) in a way that describes the mechanism of competition. May be, they then might accept my proposal that their fine assay is driven by affinity rather than avidity.

Further suggestion:
It seems that it might be more useful in the future to use the information on IgG concentration and on binding strength (1/IC 50) separately for judgement,
than to use ABAT. Why? This will help to distinguish more quickly and more easily between the roles of IgG concentrations and binding strength (affinity).

Author Response

Thank you for your very helpful comments. We agreed all comments and revised our manuscript accordingly.

Reviewer 2 Report

Takahashi et al present this manuscript that uses the multiplication of antibody titre with antibody avidity to predict a correlate of protection in vaccinated or infected patients. The idea is interesting and worthy of investigation.  I have concerns around the clarity of the presentation of the data – the whole manuscript can be made more concise and clearer for the reader. Some claims are made that cannot be traced back to the methods, or are poorly introduced (such as which patients were used to compare antibody titre/ABAT with hospitalisation and severity classification, the main drive of the ms).

The authors have blurred the lines between sections, for this kind of exploratory work the introduction should be explicit and more detailed on other attempts to find correlates of protection. The methods should be explicit and contain more information about the methods applied to generate the data. The methods contain information that should be in the introduction.

I have included further corrections and comments below for the authors to consider:

Please introduce the DCP microarray better

Line 73 – why was 1/IC50 decided on for this value?

Line 75 – the authors have decided to multiple 1/IC50 with “antibody quantity (level)”, please could you further describe what “antibody quantity (level)” means in this section?

Line 143 to 162 – this kind of text should go in the introduction, please be strict with the methods and include the information required to repeat the experiment – no long introduction or explanation is required which should be covered in the introduction.

Line 177, unless this is a methods paper (at which point the methods should be evaluated in the results with figures), there should not be a figure in the methods section. This ‘concept’ should go in the introduction

Section 2.6, please include the package used to generate data (i.e. regression analysis parameters and quality control (goodness of fit, etc))

Line 260, please adjust the X axis for Figure 3A to 300

Line 320, it is quite reaching to suggest that a transformed in vitro assay output can be used to determine the correlate of protection against hospitalisation. Please could the authors clarify is this is their intention? In general large scale clinical trials with infection rates is required to obtain data such as this. 

In addition, the authors stated that 339 patients were included that were naïve to SARS-CoV-2 – if these patients didn’t have an antibody response, how can the analysis be performed on their samples at admission in order to predict protection from hospitalisation?

Supplementary Figure S1, why is the read-out for ELISA ug/ml? Shouldn’t this value be a binding titre EC50 or AUC – a measure of a binding curve?

Author Response

Dear reviewer 2,

Thank you very much for your precise and helpful comments. We agreed all comments and revised accordingly.

Round 2

Reviewer 2 Report

I've had a look at the revised manuscript - I'm happy from my point of view that the authors have made efforts to improve the manuscript and taken into account my suggestions.